# Gaussian Process for Machine Learning-Based Fatigue Life Prediction Model under Multiaxial Stress–Strain Conditions

**DOI:** 10.3390/ma15217797

**Published:** 2022-11-04

**Authors:** Aleksander Karolczuk, Dariusz Skibicki, Łukasz Pejkowski

**Affiliations:** 1Department of Mechanics and Machine Design, Opole University of Technology, Ul. Mikołajczyka 5, 45-271 Opole, Poland; 2Faculty of Mechanical Engineering, UTP University of Science and Technology, Kaliskiego 7, 85-796 Bydgoszcz, Poland

**Keywords:** fatigue life prediction, CuZn37 brass, machine learning

## Abstract

In this paper, a new method for fatigue life prediction under multiaxial stress-strain conditions is developed. The method applies machine learning with the Gaussian process for regression to build a fatigue model. The fatigue failure mechanisms are reflected in the model by the application of the physics-based stress and strain invariants as input quantities. The application of the machine learning algorithm solved the problem of assigning an adequate parametric fatigue model to given material and loading conditions. The model was verified using the experimental data on the CuZn37 brass subjected to various cyclic loadings, including non-proportional multiaxial strain paths. The performance of the machine learning-based fatigue life prediction model is higher than the performance of the well-known parametric models.

## 1. Introduction

In the production and operation of machines, especially the means of transport, a constant effort to reduce costs and energy consumption is employed. This goal is achieved, inter alia, by weight reduction, which can be achieved by lowering safety factors, topology optimization, or by replacing traditional materials with alternative, less dense ones. Moreover, the design process departed from the infinite-life design strategy, which requires the stress to be lower than the fatigue limit. Increasingly, the machines are designed according to the safe-life design strategy for the durability estimated by the designer. When the repair is very expensive (e.g., jet engines), a damage-tolerant design strategy is applied. Here, the service of a machine with diagnosed damage was accepted [1]. Therefore, the design of machines and structures in accordance with the savings trends and the recommendations of the latest strategies requires accurate fatigue life estimation methods accompanied by an uncertainty estimation of predictions.

The drive to reduce costs also applies to the design process. For example, the use of computational models should not require expensive experimental studies to obtain additional and more reliable material data. The high expenditure especially regards the design against material fatigue because the fatigue damage process is difficult to model. First, it consists of many fatigue crack development periods, such as crack nucleation, small crack growth, and macroscopic crack growth, and these periods differ in physical damage mechanisms also influenced by size and notch effects [2]. Each period is affected by different driving forces and requires different approaches, such as the crack nucleation and crack growth approaches [3]. Second, the fatigue behavior of machines and structures generally exhibits spectacular stochastic behavior [4,5,6]. This results from the variability of fatigue loadings, parts geometry, material properties, and microstructures. Such a significant level of difficulty in fatigue assessment has led to the development of new fatigue models, despite the numerous models that have been developed in the past few decades [7,8,9,10]. These models simplified the complexity of material fatigue by using empirical or semi-empirical approaches to relate the primary loading quantities and material properties with the fatigue life. The empirical models adjust the parameters of the regression equation to the experimental data. Semi-empirical models combine fundamental physical principles and an empirical approach. Despite the use of some physical foundations to formulate models, the selected stress/strain quantities and material parameters are related by arbitrarily formulated parametric functions to obtain the best fit for the results of the experiment.

It is concluded that in the design process of engineering structures, the selection of the fatigue model is the primary problem. The choice should theoretically depend on the material properties (e.g., brittle or ductile), the type of loading (uniaxial or multiaxial, deterministic or random, proportional or non-proportional, stress ratio), and the range of deformation (elastic or elastic-plastic). In practice, owing to the multiplicity and complexity of models, the choice is most often limited by the level of the designer’s knowledge. Conversely, owing to the cost and time-consuming nature of fatigue tests, the choice also results from the limited availability of data on the material and loading.

Based on the briefly presented problems of fatigue life prediction and existing solutions, a machine learning (ML) approach is proposed as a substitute for the semi-empirical fatigue models. The main advantage of this model is that it does not require the selection of a parametric (predefined) form of the fatigue model. The fatigue ML-based model should self-accommodate existing data and correctly reflect the fatigue behavior for testing data. Among several ML approaches, the neural network (NN) is widely used [11,12,13] in the fatigue field. According to Chen and Liu [12], NN modeling requires a large number of data, and there is no standard procedure for selecting an optimal NN architecture. One of the alternative ML approaches is the Gaussian process (GP) for regression with unique features that favors its application in fatigue life prediction of materials. 

−The GP-based model requires a considerably smaller sample size of training data than other ML techniques [14,15,16]. The current analysis successfully applied a sample size of 30. It is a size comparable to the sample size for the determination of two reference stress or strain fatigue curves [17]. −The inherent feature of the GP is the estimation of the probability distribution for the model outputs [18,19]. The probabilistic output of the GP is important for its application in fatigue life prediction. Owing to this feature, a conservative design of mechanical systems can be performed.−The GP-based model for a limited dimension of the input data vector (it is considered that the maximum five-dimensional vector of input data would be necessary for life prediction, Section 2.4) is computationally very effective [20].−The GP can estimate the relevance of each component of the input vector for effective prediction [18,20]. This feature allows us to interpret the influence of the selected input quantities on fatigue life.

Existing studies on the use of GP to predict the durability of engineering machines and structures can be classified into approaches based on pure correlation of measured signals with progressive degradation of mechanical systems and hybrid approaches, including physics-based quantities.

In the first approach, the tested signals are not related to any model of the damage mechanism. Therefore, the results of this approach have mainly practical and predictive but not explanatory purposes. For example, Huchet et al. [21] used environmental parameters, such as wind speed and direction, in the fatigue assessment of wind turbine structures. In the research of Aye et al. [22] and Hong et al. [23], the prediction of the remaining life of bearings was directly based on the acquired vibration signals. Mohanty et al. [24,25] applied online signals from piezoelectric sensors attached to selected aircraft components in the GP for fatigue crack growth prediction. Hirvoas et al. [26] applied the GP to reduce uncertainties in a wind turbine numerical model considering different input data as support and blade structural properties.

Hybrid data-driven models involve input quantities related to the recognition of the failure mechanisms of a system. Słoński [27] used the GP to identify concrete properties, among others, using minimal and maximal uniaxial stress values. Hu et al. [28] applied GP to estimate the uncertainty of fatigue crack growth in turbine discs in a study on fatigue crack growth evaluation. Ling and Mahadevan [29] proposed replacing computationally expensive finite element analysis for fatigue crack growth with the GP model. Farid [30] predicted fatigue failure under stochastic loading using a stress signal at the critical section of the mechanical component. The GP model was combined with an artificial neural network to enhance the predictive performance and provide uncertainty quantification.

In the briefly reviewed papers, the GP models were adjusted and trained for a given mechanical system. Consequently, the trained models cannot be applied to other mechanical systems, and thus, they can be mostly classified as health monitoring systems [31].

Karolczuk and Słoński [32] proposed a novel approach in which the GP operates as a multiaxial fatigue life prediction model. The normal and shear stress amplitudes on the critical plane were selected as the physics-based input quantities for the GP model with the application of the squared exponential covariance function. The model was successfully verified on S355N steel and 2124 T851 aluminum alloy under the cyclic proportional combination of bending and torsion loadings at the high cyclic fatigue regime (stress-based condition). The proposed novel approach for fatigue life prediction requires further research and validation, especially under multiaxial non-proportional loading and stress–strain conditions. 

This research aims to validate the GP applied to build a multiaxial fatigue model for the life prediction of CuZn37 brass under proportional and non-proportional loadings and stress–strain conditions.

The scope of this research involved an overview of selected classic (parametric) fatigue life prediction models (Section 2), basic concepts (Appendix A), and covariance functions (Appendix B) of the GP. Next, the fatigue life was estimated with GP using different covariance functions. As a result of the calculations, the physical quantities of key importance for the fatigue process were selected and compared with the quantities indicated in the analysis of classic models. The GP results were validated by comparing the results with the results obtained using the parametric models proposed by Fatemi-Socie, Brown-Miller, Glinka et al., and Yu et al. The calculations were conducted for eight loading cases, that is, axial, torsional, combined proportional axial-torsion loading, and non-proportional loadings. The level of non-proportionality was different in the case of the applied loadings owing to the phase shift (90° out-of-phase) and various frequencies of the applied strains (four different asynchronous loadings).

## 2. Brief Review of Fatigue Life Prediction Models

Commonly applied semi-empirical fatigue models consist of fatigue damage parameters, which are scalar functions of spatial stress/strain components and reference fatigue curves. The fatigue damage parameter is used to reduce the spatial stress or strain state to a scalar quantity of the dimensions of stress, strain, or energy. It is then compared with the reference regression curves to calculate the fatigue life [33,34]. Uniaxial regression curves show an explicit relationship between the applied stress, strain, or energy values and the number of cycles to failure. These curves are the result of fatigue tests performed on a limited set of specimens and require the adoption of statistical assumptions [17]. Uncertainties in material parameters, loading and geometry can also be included in life prediction by the application of probabilistic modeling with sampling techniques [35,36,37].

The fatigue models differ in physical quantities, which were adopted as decisive factors for the fatigue process—predictors for the GP. For example, stresses, strains, loading non-proportionality factors, or strain energy, including elastic and plastic parts. These quantities, which fluctuate with time, are mostly reduced before incorporating them into the damage model to statistical parameters such as amplitudes, mean values, or maxima.

In this section, several selected models representing different physical principles are briefly described. A broader overview of the multiaxial fatigue models can be found in the papers concerning the critical plane approach [7], energy approach [38], or non-proportionality of loading [39].

### 2.1. Empirical Models

An example of a purely empirical model can be the “ellipse quadrant” proposed by Gough and Pollard for ductile materials [40,41]. Because this is an equation for a particular loading case, that is, torsion and bending, a model is a function of the applied shear $\tau_{a}$ and normal $\sigma_{a}$ stress amplitudes, as follows:(1)$${\left(\frac{\tau_{a}}{t_{-1}}\right)}^{2}+{\left(\frac{\sigma_{a}}{b_{-1}}\right)}^{2}=1,$$
where $t_{-1}$ and $b_{-1}$ are the fatigue limits for fully reversed torsion and bending, respectively. For brittle materials, Gough proposed the “ellipse arc” equation as follows:(2)$${\left(\frac{\tau_{a}}{t_{-1}}\right)}^{2}+\left(\frac{b_{-1}}{t_{-1}}-1\right){\left(\frac{\sigma_{a}}{b_{-1}}\right)}^{2}-\left(2-\frac{b_{-1}}{t_{-1}}\right)\frac{\sigma_{a}}{b_{-1}}=1.$$

The above equations define the fatigue limit state of the applied stress amplitudes. It can be developed for a state at an arbitrary number of cycles to failure, as proposed in [42]. Empirical models have limited application (only combined torsion and bending loading) because they are not consistent with invariant principles. 

### 2.2. Stress Invariants Models

The two examples presented here are attempts to adopt the Huber–Mises yield criterion for fatigue by considering the hydrostatic stress. Both are a linear combination of the amplitude of the second deviator invariant $J_{2}$, and the mean or maximum value of the hydrostatic stress $\sigma_{H}$, as in the case of the Sines [43] or Crossland [44] model, respectively: (3)$$\sqrt{J_{2,a}}+k_{s}\cdot \sigma_{H,m}=f\left(N_{f}\right),\sqrt{J_{2,a}}+k_{c}\cdot \sigma_{H,max}=f\left(N_{f}\right),$$
where $k_{s},k_{c}$ are material parameters, $f\left(N_{f}\right)$ is the reference fatigue curve. 

The fatigue models based on stress invariants are criticized mainly because of problems in their implementation of random loading and non-proportional loading [7,45]. To overcome these problems, special procedures must be implemented. 

### 2.3. Critical Plane Models

Critical plane models are based on the observation of fatigue crack formation. Based on the observation, the fatigue cracks in metallic materials nucleate and develop in certain preferred planes within the material [46,47]. Thus, the critical plane approach assumes that the stress/strain components on a specific plane are primary for fatigue crack initiation and failure. 

The stress-based Findley criterion is one of the first critical plane multiaxial fatigue models [48]. This is a linear combination of the maximum normal stress $\sigma_{n,max}$ and shear stress amplitude $\tau_{ns,a}$ on the plane for which the equation reaches its maximum, as follows:(4)$$max\left(\tau_{ns,a}+k\cdot \sigma_{n,max}\right)=f\left(N_{f}\right),$$
where k is a material parameter.

Brown and Miller [49] proposed a strain-based fatigue damage parameter composed of shear $\gamma_{ns,a}$ and normal $\varepsilon_{n,a}$ strain amplitudes on the plane experiencing the maximum shear strain amplitude. Kandil et al. [50] proposed a fatigue life prediction model based on this concept of the fatigue damage parameter, in the form
(5)$$\gamma_{ns,a}+k_{BM}\cdot \varepsilon_{n,a}=f\left(N_{f}\right),$$
where $k_{BM}$ is a material parameter.

Fatemi and Socie [51] proposed to relate the shear strain amplitude $\gamma_{ns,a}$ and maximum normal stress $\sigma_{n,max}$ normalized by the yield strength $\sigma_{yield}$, in the following form:(6)$$\gamma_{ns,a}\left(1+k\cdot \frac{\sigma_{n,max}}{\sigma_{yield}}\right)=f\left(N_{f}\right).$$

Carpinteri et al. [52] proposed a nonlinear function of amplitude and mean value of normal stress and shear stress amplitude on the critical plane related to the average principal stress directions in the following form:(7)$$\sqrt{{\left(\sigma_{n,a}+a_{CS}\sigma_{n,m}\right)}^{2}+b_{CS}\tau_{ns,a}^{2}}=f\left(N_{f}\right),$$
where $a_{CS},b_{CS}$ are material parameters. 

Papuga–Růžička [53] also proposed a nonlinear function of amplitude and mean value of normal stress and shear stress amplitude on the critical plane of its maximum, as follows
(8)$$\underset{n}{max}\left\{\sqrt{a_{PR}\tau_{ns,a}^{2}+b_{PR}\left(\sigma_{n,a}+c_{PR}\sigma_{n,m}\right)}\right\}=f\left(N_{f}\right),$$
where $a_{PR},b_{PR},c_{PR}$ are material parameters. The proposed formula was developed [45,54] to take into account the mean shear stress effect. 

Glinka, Shen, and Plumtree [55] proposed the strain energy parameter accounting for both strains and stresses in the plane of maximum shear strain, as follows:(9)$$\gamma_{ns,a}\tau_{ns,a}+\varepsilon_{n,a}\sigma_{n,a}=f\left(N_{f}\right).$$

This model was modified by Pan-Chun-Chen [56] by introducing a weighting factor $k_{G}$ to normal components, as follows:(10)$$\gamma_{ns,a}\tau_{ns,a}+k_{G}\varepsilon_{n,a}\sigma_{n,a}=f\left(N_{f}\right).$$

Ince and Glinka [57] introduced a generalized strain energy fatigue damage parameter as a function of the elastic and plastic strain energy density contributed by the normal and shear stresses and strains on the critical plane of its maximum in the following form:(11)$$\underset{n}{max}\left(\tau_{ns,a}\gamma_{ns,a}^{e}+\tau_{ns,a}\gamma_{ns,a}^{p}+\sigma_{n,max}\varepsilon_{n,a}^{e}+\sigma_{n,max}\varepsilon_{n,a}^{p}\right)=f\left(N_{f}\right),$$
where n is a unit vector that determines the orientation of the plane. 

Yu et al. [58] modified the Ince-Glinka parameter and proposed the following model:(12)$$\frac{\gamma_{ns,a}\tau_{ns,a}}{\tau_{f}^{\prime }}+\frac{2\varepsilon_{n,a}\sigma_{n,max}}{\sigma_{yield}+\sigma_{f}^{\prime }}=f\left(N_{f}\right),$$
computed on the plane of maximum shear strain, where $\tau_{f}^{\prime }$ and $\sigma_{f}^{\prime }$ are material parameters deduced from the uniaxial reference curves.

### 2.4. Summary and Model Selection

It is assumed that the GP-based fatigue model (Appendix A) can substitute any parametric functions proposed for fatigue life prediction. The substitution should be effective because the GP can map any fatigue behavior deduced from training data, and thus, the selection problem of adequate parametric function, for example, Equations (1)–(12), are omitted. The effectiveness of the GP-based model is the highest if the input data vector for the GP includes fatigue damage-related quantities. Based on the effort of many researchers and their attempt to fit the experimental data to parametric models, the primary quantities (predictors for the GP) for fatigue damage of metallic materials can be selected from the shortlist presented in Table 1. The five predictors were selected from the critical plane models as being consistent with the observed fatigue damage mechanism of metallic materials. The role of these quantities and their interaction are discussed in [59,60]. The quantities from stress-invariant models could also be considered; however, owing to the ambiguous definition of amplitudes for stress invariants [61], they were neglected in the present research. Stress-based quantities are commonly applied at high cyclic fatigue (HCF), whereas strain-based quantities are common at low cyclic fatigue (LCF). Under non-proportional loading, the principal stresses rotate, which activates the higher number of slip systems and induces the complex interaction between dislocation movements. However, applying both quantities, i.e., strain and stress, the above mechanisms are reflected in predictors and the machine learning model is able to recognize this pattern to more effectively predict the fatigue life. The application of stress-and strain-based quantities reflects any possible additional material hardening effects occurring under non-proportional loading for some metallic materials. In the present research, we consider the plane or maximum shear strain amplitude appropriate for a wide class of metallic materials with dominant shear or mixed shear/tensile damage mechanisms.

The fatigue life prediction performance of the proposed GP-based model was compared with the performance of four parametric fatigue models, including the proposal of Brown–Miller (5), Fatemi–Socie (6), Glinka et al. (8), and Yu et al. (10). The selected models required a more detailed description of the implemented reference fatigue curves $f\left(N_{f}\right)$ and weighting material parameters. The basic regression curve implemented in defining the reference curve for each model is based on the Manson–Coffin curve [62] under fully reversed cyclic torsion loading, as follows:(13)$$\gamma_{f}\left(N_{f}\right)=\frac{\tau_{f}^{\prime }}{G}{\left(2N_{f}\right)}^{b_{0}}+\gamma_{f}^{\prime }{\left(2N_{f}\right)}^{c_{0}},$$
where $\tau_{f}^{\prime }$ is the shear fatigue strength coefficient, G is the shear modulus, $b_{0}$ is the shear fatigue strength exponent, $\gamma_{f}^{\prime }$ is the shear fatigue ductility coefficient, and $c_{0}$ is the shear fatigue ductility exponent. The Brown–Miller, Fatemi–Socie, and Glinka et al. models include the weighting material parameters that can have a life-dependent form [33,63] to be fully consistent with two uniaxial strain-life curves obtained under tension-compression and torsion loadings. The torsion strain-life curve is given by Equation (13), and the tension-compression strain-life curve is
(14)$$\varepsilon_{f}\left(N_{f}\right)=\frac{\sigma_{f}^{\prime }}{E}{\left(2N_{f}\right)}^{b}+\varepsilon_{f}^{\prime }{\left(2N_{f}\right)}^{c},$$
where $\sigma_{f}^{\prime }$ is the axial fatigue strength coefficient, E is Young’s modulus, b is the axial fatigue strength exponent, $\varepsilon_{f}^{\prime }$ is the axial fatigue ductility coefficient, and c is the axial fatigue ductility exponent. The first component of Equations (13) and (14) are the elastic strain and plastic strain, respectively. Decomposition into elastic and plastic parts is consistent with the physical mechanism, and it is necessary to derive life-dependent material weighting factors appropriately. This decomposition leads to the following notations:(15)$$\varepsilon_{f}^{e}\left(N_{f}\right)=\frac{\sigma_{f}^{\prime }}{E}{\left(2N_{f}\right)}^{b},\;\varepsilon_{f}^{p}\left(N_{f}\right)=\varepsilon_{f}^{\prime }{\left(2N_{f}\right)}^{c},\;\sigma_{f}\left(N_{f}\right)=\sigma_{f}^{\prime }{\left(2N_{f}\right)}^{b},\;\tau_{f}\left(N_{f}\right)=\tau_{f}^{\prime }{\left(2N_{f}\right)}^{b_{0}},$$
where the upper indexes p and e indicate the plastic and elastic strain parts, respectively. Implementing these notions, the detailed formulas for the parametric life prediction models were obtained [63], as follows:Brown–Miller model:
(16a)$$f\left(N_{f}\right)=\gamma_{f}\left(N_{f}\right)$$
(16b)$$k_{BM}\left(N_{f}\right)=\frac{2\left[\gamma_{f}\left(N_{f}\right)-\left(1+\nu^{e}\right)\varepsilon_{f}^{e}\left(N_{f}\right)-\left(1+\nu^{p}\right)\varepsilon_{f}^{p}\left(N_{f}\right)\right]}{\left(1-\nu^{e}\right)\varepsilon_{f}^{e}\left(N_{f}\right)+\left(1-\nu^{p}\right)\varepsilon_{f}^{p}\left(N_{f}\right)},$$
where $\nu^{e}$ and $\nu^{p}$ are the elastic and plastic Poisson ratios, respectively. 

Fatemi–Socie model:


(17a)
$$f\left(N_{f}\right)=\gamma_{f}\left(N_{f}\right)$$



(17b)
$$k_{FS}\left(N_{f}\right)=\left[\frac{\gamma_{f}\left(N_{f}\right)}{(1+\nu^{e})\varepsilon_{f}^{e}\left(N_{f}\right)+\left(1+\nu^{p}\right)\varepsilon_{f}^{p}\left(N_{f}\right)}-1\right]\frac{2\sigma_{yield}}{\sigma_{f}\left(N_{f}\right)}$$


Glinka et al. model:


(18a)
$$f\left(N_{f}\right)=\gamma_{f}\left(N_{f}\right)\tau_{f}\left(N_{f}\right)$$



(18b)
$$k_{G}\left(N_{f}\right)=\frac{4\gamma_{f}\frac{\tau_{f}\left(N_{f}\right)}{\sigma_{f}\left(N_{f}\right)}-2\left(\left(1+\nu^{e}\right)\varepsilon_{f}^{e}\left(N_{f}\right)+\left(1+\nu^{p}\right)\varepsilon_{f}^{p}\left(N_{f}\right)\right)}{\left(1-\nu^{e}\right)\varepsilon_{f}^{e}\left(N_{f}\right)+\left(1-\nu^{p}\right)\varepsilon_{f}^{p}\left(N_{f}\right)}$$


Yu et al. model:


(19)
$$f\left(N_{f}\right)=\gamma_{f}^{\prime }{\left(2N_{f}\right)}^{c_{0}}+\frac{\tau_{f}^{\prime }}{G}{\left(2N_{f}\right)}^{2b_{0}}.$$


Substituting Equations (13)–(19) in (6) and (8)–(10) and implementing an iterative gradient-based procedure results in the calculation of the fatigue life $N_{cal}$ for each parametric model.

## 3. Experiment

Tubular unnotched thin-walled specimens made of CuZn37 brass were subjected to fully reversed constant-amplitude fatigue loading under various loading paths. Because the experimental test details can be found in [64], only the most important information is reported. The strain-controlled fatigue tests were conducted according to the ASTM E2207-02 standard. The failure definition was a 10% drop in the axial force or torque. The monotonic and cyclic mechanical properties of the CuZn37 brass are listed in Table 2. The regression fatigue curves (13) and (14) are shown in Figure 1. The Monte Carlo sampling technique was applied to generate the distribution of the fatigue curves ((*i*)-indexed curves in Figure 1) using a multivariate normal distribution of regression coefficients [36,65]. A sample size of 1000 was applied, estimated from the stability analysis of error indexes (Section 4.1). The blue vertical dashed lines in Figure 1 indicate the overlapping experimental fatigue life regimes of the axial and shear fatigue curves. The experimental setup included eight different loading paths with a fixed strain ratio of amplitudes $\gamma_{xy,a}$⁄$\varepsilon_{xx,a}$. Each loading path was tested using 10 specimens at different strain loading amplitudes. The shapes of the loading paths with applied strain ratios are presented in Figure 2a. The material responses in the form of stress paths are displayed in Figure 2b. The stress amplitudes implemented in the fatigue life prediction were determined for the half-life. The received experimental fatigue lives $2N_{exp}$ were within the range of [383, 191000] reversals. The experimental data file with the registered signals (strain and stress components) was uploaded to the Mendeley data repository (https://data.mendeley.com/datasets/7fbkf6y7gv/1, accessed on 25 October 2021).

## 4. Results and Discussion

The normal and shear stress/strain components on the critical plane implemented in the parametric (Section 2) and GP-based fatigue life prediction (Appendix A) models were calculated based on the registered strain and stress tensor components. Searching for the plane orientation with the maximum shear strain (the critical one), numerical simulations were conducted with the application of Euler angles, φ,θ—describing the plane orientation [66]. In the simulation, the step between subsequent values of the Euler angles was equal to 1°. For the non-proportional loading paths, instability in the calculated fatigue lives was detected as a result of the existence of different planes with the same maximum value of shear strain but with different normal stress and strain components. To overcome this problem, the critical plane was searched within the boundary $\langle 1-\delta ,1\rangle \underset{\varphi ,\theta }{\max}\gamma_{ns}$, where δ = 0.0001 was applied. Within this boundary, the maximum values of normal stress, normal strain, and shear stress were determined and applied to the fatigue life calculations (the values could be found in the file uploaded to https://data.mendeley.com/datasets/7fbkf6y7gv/1, accessed on 25 October 2021).

### 4.1. Error Indexes

The calculated fatigue lives were compared with the experimental fatigue lives to estimate the performance of each applied fatigue model. A few error indexes were implemented in which the first group belongs to purely statistical parameters of fitting, and the second group belongs to parameters to estimate the applicability of models to life prediction. The first group is based on a percentage error, defined as
(20)$$PE=\frac{logN_{exp}-logN_{cal}}{logN_{exp}}\times 100,$$
where $N_{exp}$ and $N_{cal}$ are the experimental and calculated number of cycles to failure, respectively. The statistics of PE provide information on the efficiency of the model in fatigue life prediction. The mean values, MPE and standard deviation, SD of PE were evaluated. The positive values of PE concern cases where the calculated life was shorter than the experimental one (conservative estimate), whereas non-conservative estimates were indicated by negative PE values. An MPE value equal to zero indicates a perfect prediction-unreal case owing to fatigue life scatter.

The mean and standard deviation of PE estimate the fitting properties of the applied model. However, to build (based on these parameters) the ranking of models with respect to their acceptable uncertainty in life prediction can be questionable. First, the two parameters are combined into a decisive one. Second, the values of these parameters are not intuitive if the predicted fatigue lives are acceptable. The experimental fatigue life can vary significantly even under a single loading condition [67,68]. The fatigue scatter factor is defined as:(21)$$T=\left\{\begin{array}{ccc} \frac{N_{exp}}{N_{cal}} & for & N_{exp}\geq N_{cal}\\ \frac{N_{cal}}{N_{exp}} & for & N_{exp}<N_{cal} \end{array}\right.$$

Fatigue scatter factor less than three is commonly accepted [69,70], and a value less than or equal to two exhibits a very good estimation [71,72,73]. Thus, the statistical distribution of the T factor was used to define [74] a more intuitive error index as a 0.95-quantile of the fatigue life scatter factor. The 0.95-quantile $\left(T\left(0.95\right)\right)$ was calculated based on the shape-preserving piecewise cubic interpolation of the empirical cumulative distribution of T. The value of $T\left(0.95\right)$ determines the minimum fatigue-life scatter band required to include 95% of all experimental fatigue data.

### 4.2. The Parametric Fatigue Life Prediction Models

A comparison of the experimental $2N_{exp}$ and calculated $2N_{cal}\;$fatigue lives for the four analyzed parametric models is presented in Figure 3. Each panel in Figure 3 includes the solid line of perfect life consistency enclosed by dashed lines of the scatter band T=2.0. The error bars indicate 95% prediction intervals computed using the Monte Carlo sampling technique on strain-life fatigue curves. Additionally, the results for different loading paths are marked, and the corresponding values of the error indexes are included in the legend of the figure panels. The data used for the calibration of the model, i.e., uniaxial tension–compression and pure torsion were labeled as ‘Train data’ and the remaining data as ‘Test data’. For the labeled data, the MPE, SD, and $T\left(0.95\right)$ error indexes were estimated and presented in each panel.

The error indexes are identical for the analyzed models of Fatemi–Socie, Brown–Miller, and Glinka et al., with the application of the life-dependent material parameters that equalize the performance of the parametric models under data used for calibration. The obtained error indexes MPE=−0.1%, SD=2.5%, and $T\left(0.95\right)=1.5$ for the training data and the models characterize the experimental life scatter for the CuZn37 brass under the fatigue test conditions. These values were treated in further analyses as the reference consistency of fatigue lives. Yu et al. applied constant weighting factors of shear and normal strain energy parameters, and the results for the analyzed CuZn37 brass were ineffective even under uniaxial and pure torsion loadings with MPE=−23.7% and MPE=−9.8%, respectively (Figure 3d).

The fatigue lives for the multiaxial proportional loading path were calculated using models with life-dependent material parameters within the acceptable value of error indexes (80% of the data is included within the scatter *T* = 2.0, and 100% of the data is included within the scatter *T* = 3.0). For the non-proportional loading paths, the best life prediction performance was exhibited using the Fatemi–Socie model (Figure 3a). However, the fatigue life under loading path NPR4 was unsuccessfully estimated with MPE=7.7%. The performances of the models by Brown–Miller and Glinka et al. were the worst, with MPE values exceeding 14% and 27% for the NPR4 path, respectively. The Brown–Miller model with shear strain $\gamma_{ns}$ linearly combined with the normal strain $\varepsilon_{n}$ resulted in a very conservative life estimation under non-proportional loading paths (Figure 3b). This means that the non-proportional loading paths led to a higher value of the normal strain on the critical plane of the maximum shear than expected using the Brown–Miller model. The same effect, but magnified by the shear stress on the critical plane, was observed in the model by Glinka et al. 

The tendency of higher life underestimation ($N_{cal}<N_{exp}$) for longer experimental fatigue lives was observed for all models with the life-dependent material parameters under non-proportional loading paths. It is concluded that the CuZn37 brass experienced additional hardening under non-proportional loading paths that resulted in higher stress values than expected by the models, which were calibrated by uniaxial and pure torsion loading paths. Yu et al. overestimated ($N_{cal}>N_{exp}$) the predicted fatigue lives for all loading paths.

### 4.3. The Gaussian Process-Based Fatigue Model

#### 4.3.1. Physics-Based Input Parameters (Predictors)

The GP model is assumed to substitute the semi-empirical fatigue criteria; thus, the scalar output of the logarithm of the fatigue life must be invariant under the rotation of coordinate systems. To meet this requirement, the covariance function for the GP must not directly implement the input parameters as stress and/or strain tensor components. The rotation of the coordinate system transforms the stress/strain tensor components, and thus, the model trained on the dataset valid for the original coordinate system would not apply to the rotated coordinate system. 

A state-of-the-art review of fatigue life prediction models (Section 2) indicates a few commonly applied crucial physics-based quantities. These are derived from the concept of the critical plane built on the experimental observation that fatigue cracks in metals are initiated in the plane of maximum shear stress or strain [49,75], $\tau_{ns}$, $\gamma_{ns}$. Furthermore, these microcracks could not be developed if the maximum normal stress or strain $\sigma_{n},\;\varepsilon_{n}$ on this plane were below the critical value. These four parameters are considered primary in most multiaxial fatigue life prediction models (Table 1). Owing to the zero mean stresses in the experimental data, the $\sigma_{n,m}$ predictor is omitted for components of the input data vector x, as follows:(22)$$x=\left[\gamma_{ns},\;\varepsilon_{n},\tau_{ns},\sigma_{n}\right].$$

The maximum shear strain plane was determined over the entire loading history and fixed in the fatigue life calculation process. Thus, the selected predictors (22) are invariant under the rotation of the coordinate system. Regarding the identification of multiple planes with equal maximum values of shear strain, the plane with the highest normal stress was selected as the critical plane. The relevance of each predictor was analyzed during the training process, as presented in the next section. The diagram of data flow for the fatigue life prediction based on the proposed GP fatigue model is presented in Appendix A, Figure A1.

#### 4.3.2. Training Process

The polycrystalline structure with preferred slip systems makes metallic materials sensitive to the type of fatigue loading paths. For example, the non-proportional loading characterized by the rotation of principal stresses could activate a larger number of slip systems and their intensive interactions compared to the proportional loading path [76]. These phenomena could influence the fatigue life of materials; however, this effect depends on many factors [77,78]. If the fatigue GP-based model is expected to be implemented under non-proportional loading for materials sensitive to its effect, the training data should include such a case. The analyzed experimental data included various non-proportional loading paths (Section 3, Figure 2), and the most common one (easy to replicate) with a 90° phase shift (Path NPR in Figure 1) was selected for inclusion in the training process. To analyze the effect of the non-proportional loading path on the fatigue life, two training datasets were implemented. The first training dataset includes only loading paths commonly applied to identify the parametric fatigue life prediction models (Section 2), that is, uniaxial push-pull (10 specimens) and pure torsion (10 specimens) loadings. The second training dataset additionally included the NPR path (10 specimens).

Initially, the training process included all four predictors (22), five specified covariance functions (Appendix B), and the first training dataset. The length scales $l_{i}$ for each covariance function and each predictor were used to calculate the relevance factor (*RF*), defined as:(23)$$RF={\left(\frac{l}{std\left(x\right)}\right)}^{-1},$$
where l is the length scale, and $\mathrm{std}\left(x\right)$ is the standard deviation of the analyzed predictor observed in the training data. The larger the length-scale parameter, the lower the covariance, and thus the lower the influence of its predictor. However, it also depends on the analyzed physic-based quantity; for example, the strains in the fatigue regime are several orders of magnitude below the stresses. This can be considered by normalizing the length scale using the standard deviation of the input of the analyzed predictor, $std\left(x\right)$. The reverse of $l/std\left(x\right)$ is defined as the RF. An RF value approaching zero indicates that the analyzed predictor can be ignored.

The RFs obtained for the first training dataset are shown in Figure 4. The vector of the standard deviations for the first training dataset is $\mathrm{std}\left(x\right)=\left[0.0026\left(-\right),\;0.0009\left(-\right),\;6.6\left(\mathrm{MPa}\right),\;62\left(\mathrm{MPa}\right)\right]$. Two conclusions can be drawn. First, the exponential covariance function (EX) exhibits the lowest values of the relevance factors compared to other kernels. This means that the predicted values for the test data applying the EX kernel will be characterized by a larger uncertainty than for the remaining kernels. Second, the predictor $\varepsilon_{n}$—of the normal strain on the critical plane can be neglected for M5/2, RQ, and SE kernels (near zero relevance factor). The mean percentage error (MPE), standard deviation of (PE), and $T\left(0.95\right)$ factor for all the covariance functions are in the ranges $MPE=\left[-0.07,-0.06\right]\;\%$, $SD=\left[2.0,\;2.5\right]\;\%$, $T\left(0.95\right)=\left[1.53,\;1.54\right]$. Based on the obtained results, further analysis was reduced to four covariance functions (M3/2, M5/2, RQ, and SE) and three predictors, $x=\left[\gamma_{ns},\tau_{ns},\sigma_{n}\right]$. The error-fitting indicators; MPE, SD, and $T\left(0.95\right)$ were in the same range as the initial selection. The estimated values of the hyperparameters for the analyzed kernels for the first training dataset are presented in Table 3.

The relevance factors for the second training dataset, including the non-proportional loading path NPR, are presented in Figure 5. Based on the previous analysis, estimation was conducted on four covariance functions and three predictors. Here, the vector of standard deviations of the predictors was found as $\mathrm{std}\left(x\right)=\left[0.0027\left(-\right),\;\;22.4\left(\mathrm{MPa}\right)\;124\left(\mathrm{MPa}\right)\right]$. For the second training dataset, all four kernels indicate the irrelevance of the shear stress $\tau_{ns}$ on the critical plane. Thus, in the final selection, the shear stress as a predictor was abandoned. The error fitting indicators for both selections are in the ranges of $MPE=\left[-0.07,-0.06\right]\;\%$, $SD=\left[2.15,\;2.50\right]\;\%$, $T\left(0.95\right)=\left[1.44,\;1.67\right]\;\%$.

Inclusion of the non-proportional loading path NPR into the training data increased the RFs for all kernels by approximately four times. The estimated values of the hyperparameters for the analyzed kernels for the second training dataset are presented in Table 4.

#### 4.3.3. Test Process

In contrast to the parametric models in which estimation of output uncertainty requires additional methodologies, for example, Monte Carlo sampling [36], the inherent property of the GP model is the variance estimation of the outputs. This property was utilized in the fatigue life prediction shown in Figure 6 by additional vertical bars providing 95% prediction intervals. Figure 6 presents a comparison of the experimental and calculated fatigue lives obtained by the implementation of M3/2, M5/2, RQ, and SE kernels trained on the first dataset (only tension-compression and pure torsion paths). 

The results demonstrated that within the test data (6 loading paths and 60 specimens), only fatigue lives predicted for the multiaxial proportional loading can be accepted with $MPE=\left[3.7,\;4.2\right]\%$ and all data included within the fatigue scatter band T=3.0. The fatigue lives under the non-proportional loading paths with no exception are predicted with high underestimation with $MPE=\left[22.8,\;37.6\right]\%$ and $T\left(0.95\right)=\left[13.2,\;88.4\right]$. Additionally, the 95% prediction intervals are approximately two and five times larger than those estimated for the training data with the implementation of the M3/2 and M5/2 kernels. A summary of the error indexes received for the analyzed kernels is presented in Figure 7. The lowest error index was obtained for the M3/2 kernel, but with high (95%) prediction intervals and unacceptable values of $T\left(0.95\right)>3$.

It is concluded that the fatigue GP-based model trained on the dataset without the non-proportional loading paths cannot predict the material behavior of CuZn37 brass. The non-proportional loading induced an additional fatigue phenomenon in the CuZn37 brass compared to the phenomenon induced under proportional loading.

The results obtained for the fatigue GP-based model trained on the second dataset, including one non-proportional loading path NPR, are displayed in Figure 8. By adding the results for one non-proportional loading path with 10 specimens to the training data, the prediction performance of the GP model was improved. An improvement is observed for all the applied covariance functions. The error indexes for the test dataset and analyzed kernels are practically equal (Figure 9) with $MPE=\left[-0.2,-0.1\right]\%$ and $T\left(0.95\right)=\left[2.4,\;2.5\right]$.

The training dataset with 30 specimens successfully trained the GP model for fatigue life prediction of CuZn37 brass subjected to proportional and non-proportional loading paths with test data of 50 specimens. One standard non-proportional loading path with a 90° phase shift provided sufficient information on the fatigue behavior of the GP model. Consequently, the fatigue life was successfully predicted for the four tested non-proportional loading paths (NPR1, NPR2, NPR3, NPR4) of different and complex shapes (Figure 2) and for one proportional loading path (Prop). Furthermore, the common complex measures of the non-proportional loading effect on fatigue life, such as non-proportional factors [79,80,81,82], integral approaches [66,72,82] and enclosing surface methods [83,84], are not needed.

## 5. Summary and Conclusions

The Gaussian process (GP) was applied to build the fatigue model for the life prediction of CuZn37 brass subjected to various multiaxial loading paths. The physics-based input quantities in the form of stress and strain components on the critical plane of maximum shear served as predictors in the GP-based model. These quantities are invariant under the rotation of the coordinate system, and thus, the trained fatigue GP-based model is consistent with the invariance principles. Five stationary covariance functions for the GP with length scales assigned to each predictor (anisotropic kernels) were implemented and analyzed. Two Matern class kernels (M3/2, M5/2), SE, and RQ were preliminarily accepted for final validation based on the relevance analysis of predictors. The predictive performance of the GP-based model was verified using two training datasets. The first training dataset included only uniaxial and pure torsion loading paths, and the second dataset included one non-proportional loading path. The detailed conclusions are as follows.

The relevance analysis of the applied input quantities for the fatigue GP-based model revealed that the maximum shear strain and normal stress on the plane of maximum shear are the most decisive factors for the life prediction of CuZn37 brass.The GP model trained on uniaxial and pure torsion loading paths was ineffective for the prediction of the fatigue life of CuZn37 brass under non-proportional loading paths.Two Matern-class kernels (M3/2, M5/2), the SE kernel, and the RQ kernel were successfully applied to the GP-based model with better prediction performance than the parametric commonly applied multiaxial criteria of Fatemi–Socie, Brown–Miller, Glinka et al., and Yu et al.The computational time was decreased approximately 7.8 times by applying the GP-based model compared to the parametric fatigue models.The effect of mean loading can be simply implemented in the proposed fatigue GP-based model by adding the mean components of stress/strain to the input quantities (predictors).

The GP-based model can effectively substitute parametric fatigue life prediction models if physics-based predictors consistent with invariant principles are applied. Further validation for different types of materials under the mean stress effect and a wider fatigue life regime is needed.

## Figures and Tables

**Figure 1 materials-15-07797-f001:**
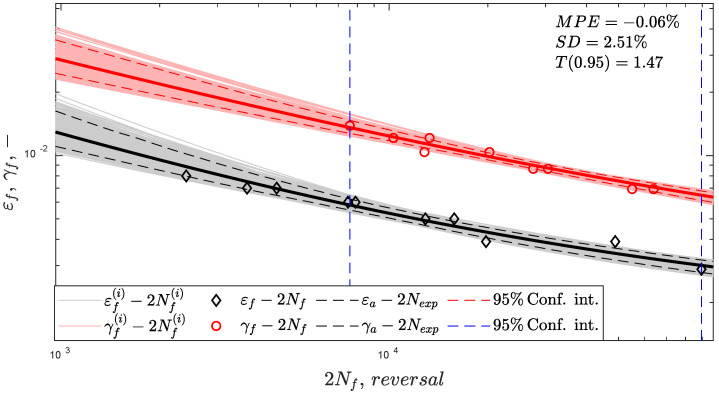
Shear and axial strain-life fatigue curves for CuZn37 with error indexes $MPE,SD,T\left(0.95\right)$ (described in Section 4.1).

**Figure 2 materials-15-07797-f002:**
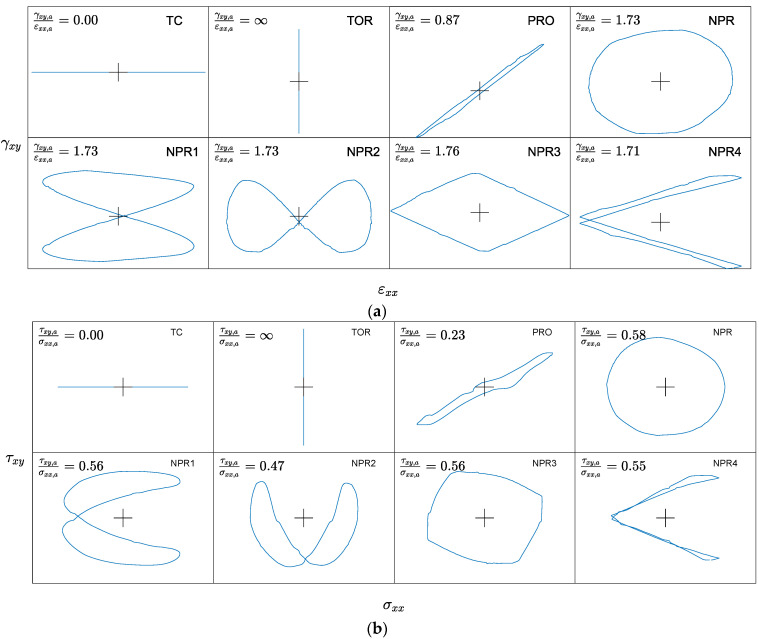
Registered experimental strain paths in (**a**) axial-shear strain space, $\varepsilon_{xx}-\gamma_{xy}$ and (**b**) axial-shear stress space, $\sigma_{xx}-\tau_{xy}$.

**Figure 3 materials-15-07797-f003:**
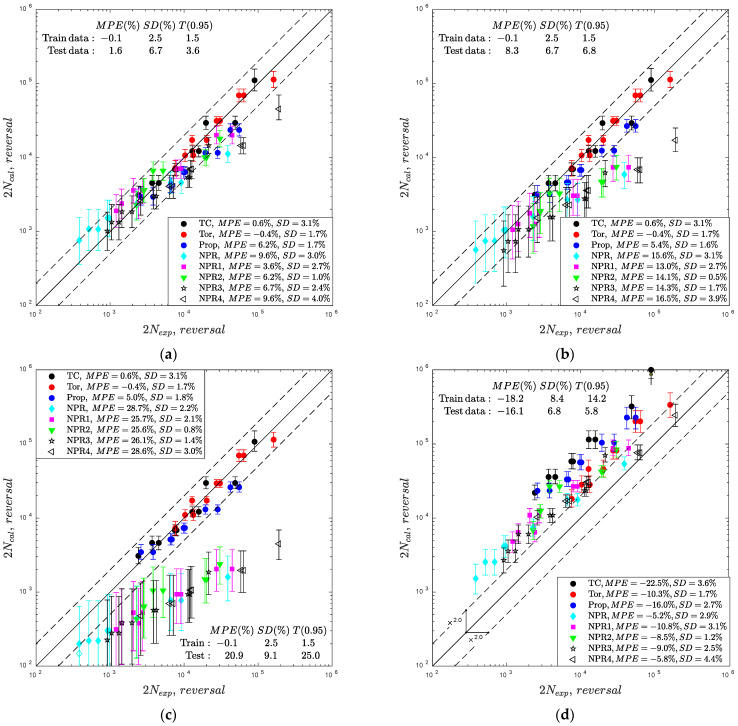
Comparison of experimental and calculated fatigue lives for the models of (**a**) Fatemi–Socie, (**b**) Brown–Miller, (**c**) Glinka et al., and (**d**) Yu et al.

**Figure 4 materials-15-07797-f004:**
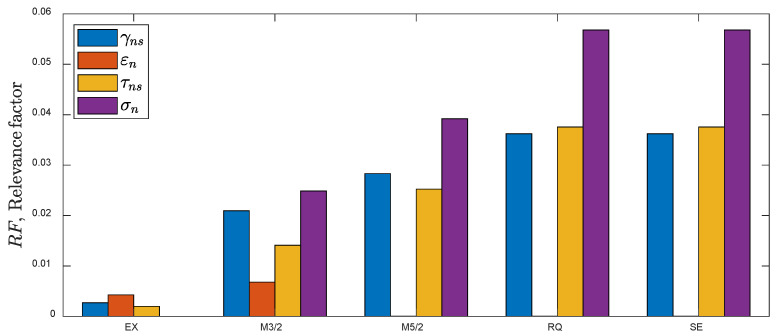
RFs obtained by application of different kernels for the first training dataset (uniaxial and torsion).

**Figure 5 materials-15-07797-f005:**
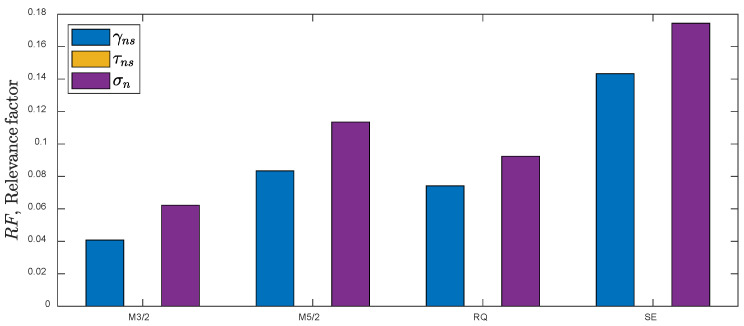
RFs obtained by application of different kernels for the second training dataset (uniaxial, torsion, and NPR path).

**Figure 6 materials-15-07797-f006:**
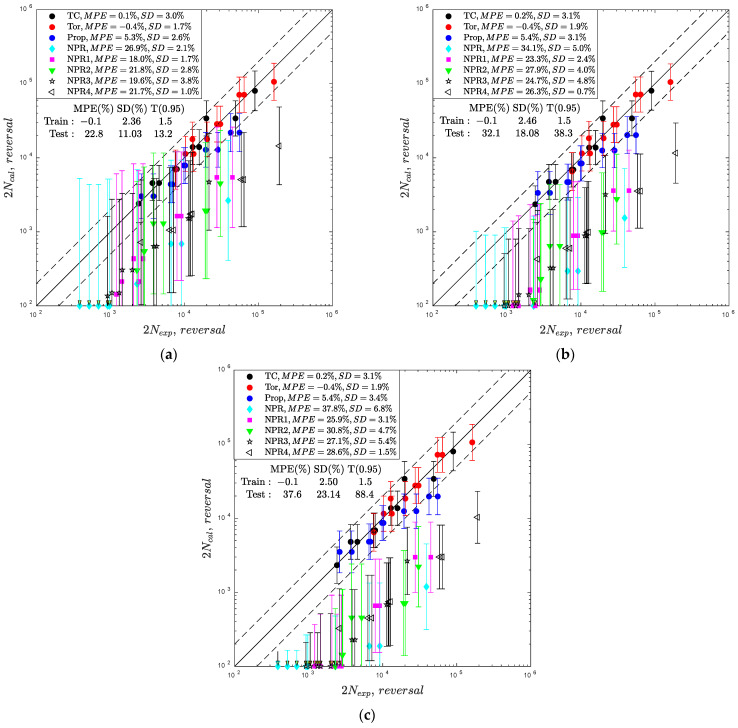
Comparison of experimental and calculated fatigue lives for the fatigue GP-based model for the first training dataset and the following covariance functions: (**a**) Matern 3/2, (**b**) Matern 5/2, (**c**) RQ/SE.

**Figure 7 materials-15-07797-f007:**
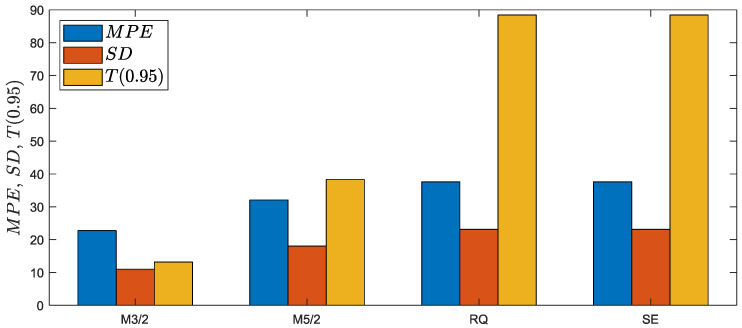
Comparison of error indexes obtained by application of different kernels to the fatigue GP-based model trained on the first training dataset.

**Figure 8 materials-15-07797-f008:**
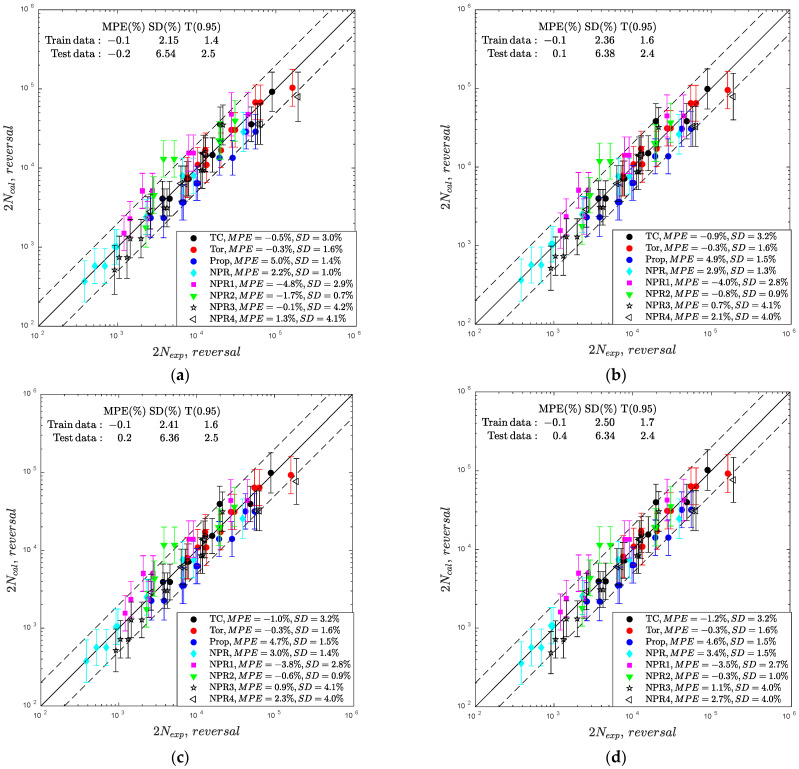
Comparison of experimental and calculated fatigue lives for the fatigue GP-based model for the second training dataset and the following covariance functions: (**a**) Matern 3/2, (**b**) Matern 5/2, (**c**) RQ, and (**d**) SE.

**Figure 9 materials-15-07797-f009:**
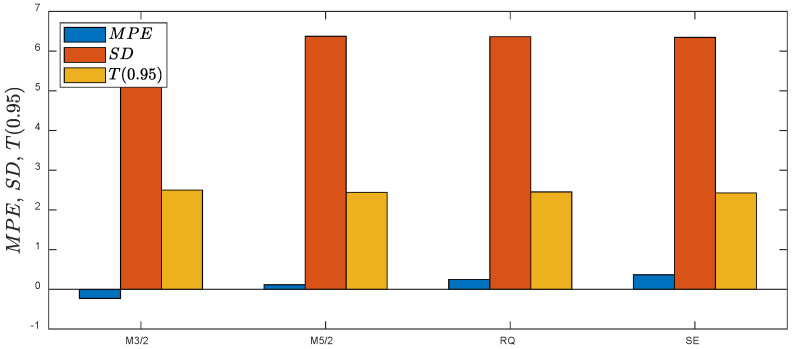
Comparison of error indexes obtained by the application of different kernels to the fatigue GP-based model trained using the second dataset.

**Table 1 materials-15-07797-t001:** Primary predictors for fatigue life of metallic materials.

No	Predictor	Description
1	$\gamma_{ns,a}$	Shear strain amplitude—primary parameter at LCF for materials with dominant micro shear cracking
2	$\tau_{ns,a}$	Shear stress amplitude—primary parameter at HCF for materials with dominant micro shear cracking
3	$\varepsilon_{n,a}$	Normal strain amplitude—primary parameter at LCF for materials with dominant tensile cracking or materials with mixed shear/tensile cracking
4	$\sigma_{n,a}$	Normal stress amplitude—primary parameter at HCF for materials with dominant tensile cracking or materials with mixed shear/tensile cracking
5	$\sigma_{n,m}$	Mean value of normal stress—primary parameter at HCF for materials with dominant tensile cracking or materials with mixed shear/tensile cracking. It reflects the beneficial effect of compressive mean stress for fatigue

**Table 2 materials-15-07797-t002:** Mechanical properties of CuZn37 brass.

***E* (GPa)**	$\nu^{e}$	$\nu^{p}$	***G* (MPa)**	$\sigma_{u}$ **(MPa)**	$\sigma_{yield}$ **(MPa)**	$K^{\prime }$ **(MPa)**	$n^{\prime }$
105	0.33	0.50	39.5	366	138	819	0.2142
$\gamma_{f}^{\prime }$	$c_{0}$	$\tau_{f}^{\prime }$ **(MPa)**	$b_{0}$	$\varepsilon_{f}^{\prime }$	c	$\sigma_{f}^{\prime }$ **(MPa)**	b
0.5065	−0.4370	204	−0.0475	0.3853	−0.5269	393	−0.0526

$K^{\prime },\;n^{\prime }$ are cyclic strength coefficient and strain hardening exponent, respectively.

**Table 3 materials-15-07797-t003:** Hyperparameters of the covariance functions for the first training dataset.

Kernel	Length-Scales			Scale-Mixture Parameter	Standard Deviation of the Noise	Standard Deviation of the (Noise-Free) Signal
	$l_{\gamma }$ **, (-)**	$l_{\tau }$ **, (MPa)**	$l_{\sigma }$ **, (MPa)**	α **, (-)**	$\sigma_{y}$ **, (-)**	$\sigma_{k}$ **, (-)**
M3/2	0.1239	480.5	2430	-	0.1099	6.74
M5/2	0.0921	262.8	1581	-	0.1106	6.12
RQ	0.0720	176.5	1091	1.527 × 10^5^	0.1109	5.75
SE	0.0720	176.5	1091	-	0.1109	5.75

**Table 4 materials-15-07797-t004:** Hyperparameters of the covariance functions for the second training dataset.

Kernel	Length Scales		Scale-Mixture Parameter	Standard Deviation of the Noise	Standard Deviation of the (Noise Free) Signal
	$l_{\gamma }$ **, (-)**	$l_{\sigma }$ **, (MPa)**	α **, (-)**	$\sigma_{y}$ **, (-)**	$\sigma_{k}$ **, (-)**
M3/2	0.0657	1996	-	0.1005	6.16
M5/2	0.0321	1094	-	0.1059	5.40
RQ	0.0361	1344	0.1301	0.1079	6.00
SE	0.0187	712.0	-	0.1096	5.83

## Data Availability

The data that support the findings of this study are openly available in the Mendeley data repository at http://doi.org/10.17632/7fbkf6y7gv.1 (accessed on 25 October 2021).

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
