# Peer review of "Gaussian Process for Machine Learning-Based Fatigue Life Prediction Model under Multiaxial Stress–Strain Conditions"

_materials, 2022, doi:10.3390/ma15217797_

Round 1
Reviewer 1 Report
Dear Author,
This manuscript presents Gussian process for machine learning-based fatigue life prediction model under multiaxial stress-strain conditions. The content of this manusript has the meaning and merits publication. However, before publishing the reviewer wants the author to amend manuscript in several points.
(1) The author insists that new point of this study is adopting Gussian process for machine learning to fatigue life prediction. What is the principal meaning of adopting the process to multiaxial fatigue life prediction? Why and how is the multiaxial non-proportional fatigue failure evaluated with high accuracy. The reviewer considers that this process is related to the microscopic plastic zone and crack extension. However, relevance to them is not discussed in the manuscript. The author needs to be more explanatory regarding why the failure life can be evaluated accurately.
(2) Introduction is not concise and how the resultant evaluation leads to the engineering design of structure and engineered parts. The reviewer considers that this method is related to the degree of strain values generated under loading and its relationship strain hardening needs to be discussed. It is related to the design stress and the resultant fatigue limit (as the limit of multiaxial fatigue failure). Amend it more properly.
(3) Phase shift and strain path controls failure life and resultant prediction via models. How does the machine learning evaluate this matter? I think the consideration regarding this point is not sufficient.
(4) The author needs to explain the meaning of parameters used in Gaussian process in machine learning to evaluate fatigue life. Strains are related; however its relationship is not made clear.
(5) Add the graph presenting the relationship between training data and basic parameters. It is essential to justify the values of this method (‘this is not the simple fitting’). It may be related to the crack extension speed and resultant fatigue life. It is needed to explain how the crack geometry could be considered into the evaluation.
(6) The reviewer considers that it is related to the strain value used as basic design of engineered parts. Underlying philosophy needs to presented (to the degree allowable).
Author Response
The comments given by the reviewers were very helpful in increasing the quality of the manuscript. All the comments were deeply analyzed and appropriate corrections were introduced to the revised version of the manuscript. The changes made in the manuscript are marked.
Detailed responses to the Reviewers’ comments are given below.
Reviewer 1
This manuscript presents Gussian process for machine learning-based fatigue life prediction model under multiaxial stress-strain conditions. The content of this manusript has the meaning and merits publication. However, before publishing the reviewer wants the author to amend manuscript in several points.
(1) The author insists that new point of this study is adopting Gussian process for machine learning to fatigue life prediction. What is the principal meaning of adopting the process to multiaxial fatigue life prediction? Why and how is the multiaxial non-proportional fatigue failure evaluated with high accuracy. The reviewer considers that this process is related to the microscopic plastic zone and crack extension. However, relevance to them is not discussed in the manuscript. The author needs to be more explanatory regarding why the failure life can be evaluated accurately.
The research aim was to “validate the GP applied to build a multiaxial fatigue model for the life prediction of CuZn37 brass under proportional and non-proportional loadings and stress-strain condi-tions.” We adopted the physical quantities applied in the various fatigue models as the input quantities to the machine learning model. In such meaning, we adopt the GP model to multiaxial life prediction, however, this phrase was not included in the manuscript. The novelty is the application of strain and stress quantities as predictors in the GP-based fatigue model. Application of applied (controlled) strains and stresses (material response) but transformed on the critical plane (invariant principle) allows the GP-based model to recognize the pattern of additional strain hardening/softening caused by e.g. non-proportional loading. Under non-proportional loading, the principal stresses rotate which activates the higher number of slip systems and induces the complex interaction between dislocation movements. But, applying both quantities, i.e. strain and stress, the above mechanisms are reflected in predictors and the machine learning model is able to recognize this pattern to more effectively predict the fatigue life. The appropriate additional comments were added to subsection 2.4.
(2) Introduction is not concise and how the resultant evaluation leads to the engineering design of structure and engineered parts. The reviewer considers that this method is related to the degree of strain values generated under loading and its relationship strain hardening needs to be discussed. It is related to the design stress and the resultant fatigue limit (as the limit of multiaxial fatigue failure). Amend it more properly.
The presented research proves that the properly trained machine learning model, here the Gaussian process, can effectively predict fatigue life under the multiaxial strain-stress condition. It means that in the design process, the crucial step of selecting the fatigue model can be substituted by the machine learning model, e.g. Gaussian process (see Introduction lines 54-66). The machine learning model contrary to the classic (parametric) model automatically adapts to existing data reducing the prediction error to a minimum. Strain hardening was discussed in the previous answers and some additional comments were added to subsection 2.4.
(3) Phase shift and strain path controls failure life and resultant prediction via models. How does the machine learning evaluate this matter? I think the consideration regarding this point is not sufficient.
This problem was discussed in point 1.
(4) The author needs to explain the meaning of parameters used in Gaussian process in machine learning to evaluate fatigue life. Strains are related; however its relationship is not made clear.
Four quantities, two stress, and two strain were selected based on analysis of semi-empirical fatigue models. Their selection was based on an analysis of the model of the physical process of crack initiation and development. Their choice did not assume any mathematical relationship between these quantities. As noted in the introduction, the stress/strain quantities and material parameters in semi-empirical fatigue models are related by arbitrary functions, guaranteeing the best fit to the experiment results. The only expertise provided to the applied machine-learning method is the choice of stress and strain quantities. The problem of the interaction of shear and normal strain/stress components on the critical plane and their influence on fatigue life is still discussed in many articles (Gates, N.R.; Fatemi, A. On the consideration of normal and shear stress interaction in multiaxial fatigue damage analysis. Int. J. Fatigue 2017, 100, 322–336. and Gates, N.; Fatemi, A. Friction and roughness induced closure effects on shear-mode crack growth and branching mechanisms. Int. J. Fatigue 2016, 92, 442–458.). In general, the shear components induce (or reflect) the dislocation movements and formation of plastic slip bands. The influence of normal components is less clear, it is observed that the positive normal stress induces a crack opening mechanism, reduces friction between crack planes, and influences the crack branching phenomena. The mentioned papers were added to the reference list and cited in subsection 2.4.
(5) Add the graph presenting the relationship between training data and basic parameters. It is essential to justify the values of this method (‘this is not the simple fitting’). It may be related to the crack extension speed and resultant fatigue life. It is needed to explain how the crack geometry could be considered into the evaluation.
The increasing interest in machine learning models is evident. This phenomenon is based on highly developed and verified computation algorithms that are ready to be implemented in various engineering problems. The Gaussian process is one of the machine learning methods with unique features that favor its application in fatigue life prediction of materials (see Introduction). The current paper presents one of the first research on the application of the Gaussian process for fatigue life prediction. The proposed method must be verified on many materials and loading conditions. Different covariance functions and different predictors must be analyzed and verified. The proposed models should be interpretable to increase trust in the prediction. The current papers will not solve these problems. However, the application of physics-based predictors with the invariant principle is a valuable step in developing the model for fatigue life prediction. We attend to future work to introduce into the model the additional physics constraints to make the model more interpretable.
The machine learning models have advantages over the human-proposed parametric models in recognizing the pattern in multidimensional data. Most of the existing parametric fatigue models are based on two input quantities, but the fatigue problem is more complex. In the current research, we applied four predictors. Visualizing such several quantities in relation to any other variables is not possible. There are special tools developed for machine learning models to visualize some relations but it is done only partially to one or two variables.
We predict fatigue life in the range up to cycles. Thus, we assume that the period of macro-crack development is negligibly small relative to fatigue life. Therefore, the present model is not based on fracture mechanics. Such quantities as the crack extension speed and geometry are not considered. However, considering the proposed model as a substitution for classic fatigue models we can apply for example the critical distance theory (Susmel, L.; Taylor, D. A critical distance/plane method to estimate finite life of notched components under variable amplitude uniaxial/multiaxial fatigue loading. Int. J. Fatigue 2012, 38, 7–24.) to take into account different notch geometry. But, to stay on the current research aim and not distract the readers this issue was not discussed in the manuscript.
(6) The reviewer considers that it is related to the strain value used as basic design of engineered parts. Underlying philosophy needs to presented (to the degree allowable).
Indeed, there are fatigue models based solely on strains, such as the Itoh model. But our proposal falls into the category of strain-stress models. We recognize that the contribution of stress and strain and their interaction is essential, as the Fatemi-Socie model, for instance, assumes it. Justification for the choice of quantities that are input parameters (predictors), both strain and stress, can be found in Section 2.4 and Section 4.3.1.
Reviewer 2 Report
The manuscript describes a new method for fatigue life prediction under multiaxial stress-strain conditions based on machine learning. The Gaussian process for regression is used to build a fatigue model. The paper is well written and should be very interesting to readers dealing with lifetime predictions in the low-cycle fatigue domain of metallic materials. The manuscript has got important aspects of novelty and of high scientific level. However, some minor revisions are necessary prior to publishing in the journal.
Reviewer’s remarks:
1) The review of critical plane models should be improved. First, authors describes a stress-based model of Findley. Then, energy-based formulas are provided. Finally, Carpinteri and Papuga/Ruzicka fatigue criteria, which are stress based, are mentioned. This is not a logical order. It is recommended to move the description of last two criteria after Findley criterion.
2) There are also newer fatigue parameters for prediction in HCF. For instance Papuga and co-authors proposed some models having better predictions taking into account the mean stress effect. At least, mention it in the section 2.3 or in the introduction.
3) Experimental strain paths are shown in Fig.2. Please, provide similar graphs of stress responses in a new figure.
4) Page 17 – caption of Fig.8 is on a new page, which is not acceptable.
Author Response
Response to Reviewers
The comments given by the reviewers were very helpful in increasing the quality of the manuscript. All the comments were deeply analyzed and appropriate corrections were introduced to the revised version of the manuscript. The changes made in the manuscript are marked.
Detailed responses to the Reviewers’ comments are given below.
Reviewer 2
The manuscript describes a new method for fatigue life prediction under multiaxial stress-strain conditions based on machine learning. The Gaussian process for regression is used to build a fatigue model. The paper is well written and should be very interesting to readers dealing with lifetime predictions in the low-cycle fatigue domain of metallic materials. The manuscript has got important aspects of novelty and of high scientific level. However, some minor revisions are necessary prior to publishing in the journal.
Reviewer’s remarks:
1) The review of critical plane models should be improved. First, authors describes a stress-based model of Findley. Then, energy-based formulas are provided. Finally, Carpinteri and Papuga/Ruzicka fatigue criteria, which are stress based, are mentioned. This is not a logical order. It is recommended to move the description of last two criteria after Findley criterion.
Corrected.
2) There are also newer fatigue parameters for prediction in HCF. For instance Papuga and co-authors proposed some models having better predictions taking into account the mean stress effect. At least, mention it in the section 2.3 or in the introduction.
Added to subsection 2.3.
3) Experimental strain paths are shown in Fig.2. Please, provide similar graphs of stress responses in a new figure.
The figure was added.
4) Page 17 – caption of Fig.8 is on a new page, which is not acceptable.
We do our best to properly edit the manuscript.
Reviewer 3 Report
The manuscript "Gaussian process for machine learning-based fatigue life prediction model under multiaxial stress-strain conditions" has been reviewed. It deals with an interesting novel method for the prediction of fatigue life under multiaxial stress-strain conditions.
The manuscript is clear and well organized, english acceptable.
In my opinion in can be accepted after the following minor revision:
Ref. 45, 62, 72 are not called in the main text.
Author Response
Response to Reviewers
The comments given by the reviewers were very helpful in increasing the quality of the manuscript. All the comments were deeply analyzed and appropriate corrections were introduced to the revised version of the manuscript. The changes made in the manuscript are marked.
Detailed responses to the Reviewers’ comments are given below.
Reviewer 3
The manuscript "Gaussian process for machine learning-based fatigue life prediction model under multiaxial stress-strain conditions" has been reviewed. It deals with an interesting novel method for the prediction of fatigue life under multiaxial stress-strain conditions.
The manuscript is clear and well organized, english acceptable.
In my opinion in can be accepted after the following minor revision:
Ref. 45, 62, 72 are not called in the main text.
The Mendeley reference manager was applied. The reference list was revised.
Round 2
Reviewer 1 Report
Dear Authors,
This manuscript presents Gaussian process for machine learning-based fatigue life prediction model under multiaxial stress-strain conditions.
The revised manuscript gives sufficient background for use of GP process for machine learning-based fatigue life prediction under multiaxial loading.
The relationship between parameters and predictors is also mentioned sufficiently.
This manuscript can be accepted in the reviewer's opinion.